# Interaction between Polycomb and SSX Proteins in Pericentromeric Heterochromatin Function and Its Implication in Cancer

**DOI:** 10.3390/cells9010226

**Published:** 2020-01-16

**Authors:** Simone Johansen, Morten Frier Gjerstorff

**Affiliations:** 1Department of Cancer and Inflammation Research, Institute for Molecular Medicine, University of Southern Denmark, 5000 Odense, Denmark; sijohansen@health.sdu.dk; 2Department of Oncology, Odense University Hospital, 5000 Odense, Denmark; 3Academy of Geriatric Cancer Research (AgeCare), Odense University Hospital, 5000 Odense, Denmark

**Keywords:** polycomb protein, Pericentromeric heterochromatin, SSX, genomic instability

## Abstract

The stability of pericentromeric heterochromatin is maintained by repressive epigenetic control mechanisms, and failure to maintain this stability may cause severe diseases such as immune deficiency and cancer. Thus, deeper insight into the epigenetic regulation and deregulation of pericentromeric heterochromatin is of high priority. We and others have recently demonstrated that pericentromeric heterochromatin domains are often epigenetically reprogrammed by Polycomb proteins in premalignant and malignant cells to form large subnuclear structures known as Polycomb bodies. This may affect the regulation and stability of pericentromeric heterochromatin domains and/or the distribution of Polycomb factors to support tumorigeneses. Importantly, Polycomb bodies in cancer cells may be targeted by the cancer/testis-related SSX proteins to cause derepression and genomic instability of pericentromeric heterochromatin. This review will discuss the interplay between SSX and Polycomb factors in the repression and stability of pericentromeric heterochromatin and its possible implications for tumor biology.

## 1. Structure and Function of Polycomb Protein Complexes

Polycomb-group (PcG) proteins have been the subject of great scientific interest since it was discovered that they repressed the expression of *Drosophila* HOX genes and thus played a role in normal *Drosophila* development [1,2]. It is now well established that PcG proteins negatively regulate expression of genes essential for embryonic development, stem cell renewal and multicellular differentiation in *Drosophila* and vertebrates [3,4,5,6], hence alterations of PcG protein function have been linked to the development of a diverse range of cancers [7,8]. 

The PcG family comprises a diverse set of proteins that assemble into transcriptional-repressive complexes, called PcG repressive complexes (PRCs), that epigenetically modify chromatin. Two major PRCs have been described, namely PRC1 and PRC2 [9,10]. The four core subunits of canonical PRC1 (cPRC1) include an E3 ubiquitin ligase (RING1A/B), as well as proteins of the chromobox (CBX2/4/6/7/8), polyhomeotic-like protein (PCH1–3) and PcG group ring finger protein (PCGF2/4) families, respectively. In contrast, non-canonical PRC1 (ncPRC1) contains PCGF1/3/5/6 and lacks the CBX domain and instead possesses the zinc-finger domain containing RYBP or YAF2 protein [2,9,10,11]. PRC2 comprises the methyltransferase enhancer of zeste homologue 2 (EZH2), the enzymatic activity of which depends on the binding of two other core subunits, embryonic ectoderm development (EED) and suppressor or zeste 12 (SUZ12). Two other PRC2 have been identified; PRC2.1 with Tudor-domain containing PCL1/2/3 proteins, and PRC2.2 containing the accessory zinc-finger proteins AEBP2 and JARID2 [2,9,10,12]. 

Both types of PRC complexes are able to modify histones, specifically the tri-methylation of lysine 27 on histone 3 (H3K27me3) and the mono-ubiquitination of lysine 119 on histone 2A (H2AK119ub) by PRC2 and PRC1, respectively [13,14]. Initially, PRC recruitment was proposed to be mediated in a hierarchical manner, in which the deposition of H3K27me3 by PRC2 recruited cPRC1 [2,14]. However, the combinatorial assembly of mammalian PRC1 subunit homologues gives rise to various functionally distinct ncPRC1 complexes that can occur and function independent of PRC2 and additionally can recruit PRC2 [11,15,16,17], suggesting that PcG recruitment to chromatin and gene silencing is more complicated and complex than first assumed. 

## 2. Genomic Targeting of PcG Complexes

The genome targeting of PRC has been investigated intensively. The *Drosophila* PRC complex, PhoRC, contains the sequence-specific DNA-binding protein Pleiohomeotic (Pho) that binds PcG response elements (PRE) at PcG targets in order to recruit and stabilize PRC binding to chromatin genome-wide [18,19]. In contrast, the mammalian Pho homologue Ying Yang1 (YY1) binds active promoters but does not bind most PcG targets [18]. Additionally, cPRC and some ncPRC1 complexes do not possess DNA-binding activity [10], raising the intriguing question of how PcG are targeted to chromatin. Several studies have revealed a prominent correlation between PcG proteins and hypomethylated CpG islands (CGIs) [20,21,22], suggesting that CGIs act as a mammalian PRE. Specifically, the DNA-binding histone demethylase KDM2B recruits ncPRC1.1 to hypomethylated CGIs by stably associating with ncPRC1 [9] and recognizing and binding non-methylated CGIs through its zinc-finger CxxC domain [2,22]. 

Several other DNA binding proteins have been shown to be involved in recruiting PcG, but whether these DNA-binding factors act directly or indirectly is unclear. It is more likely that genomic targeting of PcG is controlled by various factors, such as non-coding RNAs (ncRNAs), chromatin-protein interactions, co-factors and chromatin modifications, that collectively regulate a complex and context-dependent PcG machinery [2,23]. For instance, the protein RYBP or its paralog YAF2 interact with core protein RING1B of ncPRC1 in order to recruit ncPRC1 [11]. PRC2.2 associates with JARID2 and AEBP that recruit PRC2.2 to chromatin by recognizing and binding H2AK119ub and simuntaneously stimulates the catalytic activity of core subunit EZH2 and thus facilitating H3K27me3 deposition [12]. This example illustrates the delicate interplay between chromatin modification and trans-acting proteins. Moreover, JARID2 is also implicated in targeting PRC2.2 to the tandem repeat A or B and F of the long-ncRNA (lncRNA) *Xist* which, in cooperation with ATRX, induces X-chromosome inactivation during mammalian dosage compensation [24,25]. HOTAIR and KCNQ1, two other lncRNAs, target PcG proteins to the *HOX* and mouse *Kcnq1ot1* loci, respectively [26], and short abortive RNAs transcribed from the 5′-end of PcG target genes help transcriptional repression by binding of PRC2 in cis [27]. However, few cases are reported in which specific nascent RNAs inhibit PRC2 binding or its catalytic activity [28,29], suggesting different roles of RNAs in genomic targeting of PcG.

Thus, genome targeting of PcGs is highly complex and remains an important topic of research. Further investigation of PcG targeting will elucidate how PcG proteins regulate gene expression at a molecular level, and might contribute to our understanding of how perturbation of PcG may lead to diseases such as cancer. 

## 3. The Role of PcG Complexes in Repression of Pericentromeric Heterochromatin

How the binding of the PRC complexes and the deposition of H3K27me3 and/or H2AK119ub onto chromatin promote gene silencing is still unclear. Several mechanisms have been suggested, including directly impairing transcriptional elongation [30,31,32,33] or compacting chromatin, which eventually hinders transcription. Chromatin compaction can be achieved by binding and compaction of adjacent nucleosomes [34,35,36,37] and/or long range chromatin looping, creating large repressed chromatin domains [2,38], or by recruiting chromatin remodeling complexes facilitating heterochromatin or inhibiting those facilitating euchromatin [39,40,41].

Although PcG proteins are generally considered regulators of facultative chromatin, thus regulating gene expression, PcG proteins have also been demonstrated to be involved in the formation of constitutive heterochromatin [16,42,43,44,45]. Constitutive heterochromatin is found mainly at the centromeric and telomeric regions of chromosomes, which are gene-poor regions crucial for maintaining structural organization of chromosomes and genomic integrity. Constitutive heterochromatin is generally comprised of tandem DNA repeats that are kept repressed by DNA methylation in conjunction with repressive factors and marks such as HP1 and H3K9me2/3 [46]. The formation of constitutive heterochromatin is initiated by the tri-methylation of H3K9 by the histone methyltransferases SUV39H1 and SUV39H2 and the subsequent binding of heterochromatin protein 1 (HP1) isoforms to H3K9me3. Moreover, constitutive heterochromatin is often heavy methylated at CGIs, which is mediated by the three DNA methyltransferases (DNMT) DNMT1, DNMT2 and DNMT3. HP1 and methylated CGIs both function as docking sites for additional factors that promote transcriptional silencing, DNA methylation and chromatin compaction, which are important for the formation of constitutive heterochromatin [46,47,48].

Pericentromeric heterochromatin (PCH) is comprised of satellite II and III DNA repeats and each PCH domain comprises hundreds of thousands of base pairs. Recent studies suggest that PCH may support centromere function and might be essential for architectural and topological organization of the nuclear department [49]. PCH is kept in an inactive state by the continuous repression facilitated by the SUV39H/H3K9me3/HP1/DNMT pathway. However, in specific developmental settings [16,42,43,44] or in malignancies [45,50,51,52], hypomethylation of PCH allows the deposition of PcG. In early mouse preimplantation embryos, maternal PCH is enriched in H3K9me3, HP1 and H4K20me3 mediated by SUV39H1/2 and SUV420H1/2. To the contrary, paternal PCH is initially devoid of SUV39h-mediated H3K9me3, and is dependent on the incorporation of maternal PRC1 and the subsequent recruitment of PRC2 and H3K27me3 deposition to ensure chromatin compaction [43,53,54]. This parent of origin-dependent PCH repression supports the concept that PRC recruitment is inhibited by H3K9me3/HP1, as PRC is depleted from maternal, but not paternal, PCH. 

Several studies have investigated the relationship between H3K9me3, CGI methylation and PcG recruitment to PCH in human and mice embryonic stem cells [16,42,43,44,54], and found that PRC2 recruitment correlates with loss of methylation. In this regard, it has been suggested that H3K9me3/HP1 and methylated CGI antagonize PRC recruitment, and thus inhibition of SUV39H/DNMT and complete loss of H3K9me3/HP1 and CGI methylation allows PRC recruitment and activity. In accordance with the observation that KDM2B recruits ncPRC1.1 to hypomethylated CGIs [22], it might be that complete loss of CGI methylation permits KDM2B/ncPRC1.1 recruitment, and hence H2Ak119ub deposition and PRC2 recruitment at PCH [16,42]. PRC2 can be recruited to PCH in the absence of RING1B/H2AK119ub [42], which might be achieved through interaction with BEND3 and the NuRD complex. BEND3 is enriched at PCH upon DNA hypomethylation and recruits the MBD3/NuRD complex that favors PRC2 recruitment, possibly by deacetylating H3K27 [46,55]. 

Thus, regardless of how PRC is recruited, demethylation of H3K9 and CGIs seems to set the stage for PRC deposition on PCH. In cancers, satellite DNA of PCH is often demethylated [56,57,58], which in many cases leads to unfolding and transcriptional derepression of PCH, and thus PCH instability. Since PCH instability has been linked to the pathogenesis of multiple malignancies [59,60,61,62,63] and as PcG complex deposition might serve as a compensatory mechanism to repress PCH and possibly balance genomic stability upon global hypomethylation, it is of great importance to decipher the underlying mechanisms.

## 4. PcG Bodies in Premalignant and Malignant Cells

In malignant cells, PcG proteins are found in relatively large (0.5–1 µm) nuclear aggregates, referred to as PcG bodies [45,51]. These structures are much larger than the widespread chromatin-associated PcG foci, which can be observed in most cells. PcG bodies appear frequently in melanoma cells but can be found in many types of cancer [45,52] (Figure 1). We have further demonstrated that premalignant lesions in the form nevi also frequently contain PcG bodies. In agreement, immortalized melanocytes contained small, but distinct, PcG aggregates [52]. Interestingly, recent data has demonstrated that PcG proteins condensate to form PcG aggregates though phase seperation [64,65]. 

PcG bodies have been demonstrated to be associated with the megabase PCH region located on 1q12 and with similar PCH structures on chromosome 9 and 10 [45,50,51]. PCH is prone to breakage, as exemplified by the 1q12 PCH domain, which is often involved in translocations and duplications of the 1q arm in cancer [66] and Immunodeficiency, Centromeric instability, and Facial anomalies syndrome (ICF) [67,68]. These abnormalities are associated with PCH unfolding and transcriptional derepression of PCH, which in many cases appear to be tightly linked to loss of DNA methylation. For ICF, this is commonly caused by inactivating mutations in the gene encoding DNMT3B [69,70], but in cancer, the mechanism remains elusive. Loss of DNA methylation and repressive factors such as HP1 and H3K9Me2/3 that collectively repress PCH in normal cells may generate nucleation sites for deposition of PcG proteins. In agreement, we demonstrated that accumulation of PcG proteins on 1q12 PCH domains correlates with loss of methylation during melanoma progression, and that inhibition of DNMTs could induce the formation of PcG bodies [52]. PcG complexes have also been demonstrated to deposit on hypomethylated CGIs [20,21,22], which are structurally different from the satellite II and III repeats of PCH. These data suggest that demethylation may be a primer for PcG body formation, and future studies should address whether loss of DNMT activity is involved.

Understanding of the role of PcG bodies in cancer cell biology remains an important subject of investigation. It has been suggested that PCH function as molecular sinks, depleting PcG factors from other domains and thereby deregulating gene silencing [50]. There may also be cancer-related effects of deregulating PCH itself. The deposition of PcG complexes may serve as a compensatory mechanism to repress PCH and preserve genomic stability upon global hypomethylation. PcG bodies may also be important for preventing cells from undergoing senescence in response to oncogene expression and increased proliferation. PcG bodies are present in nevi, which express oncogenes such as BRAF and NRAS [71,72], and we have demonstrated that PcG bodies emerge in SV40ER- and hTERT-immortalized melanocytes [52]. Since unfolding and depression of satellite DNA has been observed in several types of senescence [73] and may be mechanistically implicated in the senescence response, installation of PcG factors on these domains may prevent a senescence response in premalignant cells and increase proliferative capacity. Further studies of the function of PcG bodies will be important for understanding the role PCH in homeostasis and disease.

## 5. The SSX Family of Chromatin-Modulating Proteins

The Synovial Sarcoma, X-breakpoint (SSX) family comprises six highly similar members with additional splicing variants (i.e., SSX1, SSX2, SSX3, SSX4, SSX5 and SSX7) and several pseudogenes [75]. The proteins are strictly expressed in the spermatogonia of testis in healthy individuals, but ectopic expression is found in many different types of human cancer [76,77]. The proteins are most frequently detected in melanoma, where approximately 40% of tumors are positive for SSX2, SSX3 or SSX4 [78]. These characteristics, and the fact that SSX proteins are subject to T-cell responses in cancer patients, makes them members of the cancer/testis antigen group [79]. SSX proteins and other cancer/testis antigens have been explored as targets for therapeutic cancer vaccines and T-cell therapy over the last decades [79]. 

All SSX proteins contain two conserved domains, the Krüppel-associated box (KRAB) and SSX repression domain (SSXRD), both of which have been demonstrated to suppress gene expression in reporter assays [80,81]. The former is a classical protein-binding domain, although it only expresses low similarity with canonical KRAB domains and does not interact with KAP1 [82]. Instead, the KRAB domain of SSX has been demonstrated to interact with SSX2IP and RAB3IP [83], but the consequences of these potential interactions remains unexplored. We have recently reported that the SSXRD structurally resembles a C2H2 binding zinc finger domain, which is one of the most common eukaryotic DNA binding domains and is frequently found in transcription factors [74]. SSX proteins contain a short alpha-helical structure similar to the one that mediates binding to the major groove of DNA in C2H2 zinc finger motifs (C2H2-ZNF) [84]. This is in agreement with the DNA-binding properties of SSX2 [85] and the importance of the SSXRD for the chromatin-association of SSX2 [74]. Thus, the composition of SSX proteins resembles that of the ∼400 known KRAB zinc finger proteins [86] and suggesst that SSX proteins may retain a similar modular function, where the Zinc finger domain mediates DNA binding and the KRAB domain mediates recruitment of repressive chromatin modulating factors (Figure 2). Further studies should be done to delineate the DNA-binding properties of SSX molecules. 

The SSX genes were first determined to be involved in the interchromosomal rearrangement t(X;18)(p11.2; q11.2) by producing the SS18/SYT-SSX fusion oncogene, which is observed in nearly 100% of synovial sarcoma tumors. In this fusion product, eight C-terminal amino acids of SS18 are replaced by 78 amino acids of the C-terminus of SSX1, SSX2, or SSX4 [87]. Importantly, SS18-SSX is an important oncogenic driver of synovial sarcoma, and recent studies have revealed that it causes deregulation of developmental programs to drive cellular transformation [88,89]. SS18 is a component of the SWI/SNF chromatin remodeling subcomplexes, and the oncogenic property of SS18-SSX is at least partially based on its ability to functionally change and hijack BAF complexes to PcG chromatin domains [88,90]. SS18-SSX was further demonstrated to bind the ncPRC1.1 protein KDM2B and thereby redirect SWI/SNF complexes to hypomethylated CpG islands, consequently causing deregulation of developmental programs to drive transformation [89]. Similar to our results presented above [74], these studies suggest that the SSXRD is important for chromatin-targeting of SS18-SSX.

Little is known about the cellular functions of SSX proteins, but we have demonstrated that SSX proteins support the growth of melanoma cells [91]. The importance of SSX proteins in cancer cell proliferation was further validated by other studies showing that SSX proteins activate several important mitogenic pathways, such as MAPK and Wnt [92]. Interestingly, results from our group showed that SSX2 induced senescence with classical features including enlargement of the cytoplasm, cell growth arrest, enhanced B-galactosidase activity and DNA double strand breaks, in different types of cells [91]. Senescence can be induced in response to overexpression of oncogenes, like BRAF and KRAS, without cooperating genetic alterations [93,94,95,96]. The role of SSX2 in supporting cell proliferation and its ability to induce senescence suggests that SSX proteins may have oncogenic potential. Further studies should be conducted to clarify the specific role of oncogenesis.

## 6. Interaction between SSX and PcG Factors

A functional link between SSX molecules and PcG proteins has been proposed in multiple studies. In several types of malignant cells, the localization of SSX and SS18-SSX overlaps with a component of the PRC1 complex [74,85,97,98], including BMI, EZH1, RING1A and RING1B. Colocalization with the PRC2 factor EZH2 has also been reported [85]. As mentioned, PcG proteins catalyze the histone modifications H3K27me3 and H2AK119ub, which are instrumental for their repressive function. We found that the nuclear distribution of wild-type SSX2 was identical to H2AK119ub, whereas only limited overlap with H3K27me3 was observed. Interestingly, we found that SSX2 were coexpressed with BMI1 and EZH2 in the spermatogonia of the testis, but were inversely correlated with the H3K27me3 modification, suggesting that SSX antagonizes PcG function [85]. Colocalization of SSX and PcG proteins seems to be predominant in PcG bodies, but a more genome-wide association may be the case. The latter is supported by ChIP-seq studies of the chromatin-association of SS18-SSX, which show predominant association of this protein with H3K27me3 and PcG domains [88,99]. The genome binding profile for wild-type SSX remains uncharacterized. At present, the molecular interaction between SSX molecules and PcG factors remains controversial. Although there is a clear association between the two, it is not known how SSX molecules are recruited to PcG domains. Several studies have demonstrated that SS18-SSX copurifies with various PcG factors [100,101], but similar results have not been obtained with the wild-type SSX proteins. We investigated this using gel filtration of nuclear extracts and found no overlap between SSX2 and PRC1 and PRC2 fractions [85]. On the other hand, interaction between SS18-SSX and the ncPRC1.1 subunit KDM2B is dependent on the presence of the SSXRD in the fusion protein, and wild-type SSX alone immunoprecipitates with KDM2B [89]. This suggests that the SSXRD may mediate binding of SSX proteins to at least the PRC1.1. The SSXRD also seems to mediate the association of wild-type SSX with PcG bodies and with chromatin in general [74]. The zinc finger-like structure of this domain suggests a direct interaction with specific DNA/chromatin that may be guided by specific target sequences, co-factors or chromatin modifications that may overlap with deposition of PcG factors. 

## 7. SSX-Mediated Derepression of PcG-Silenced Heterochromatin

Recently, we found that SSX proteins deplete PcG bodies in cancer cells and induce genomic instability [85,91]. This was an important observation, since 1q12 PCH is one of the most frequent sites of genomic breakage in cancer and is linked to the pathogenesis of this disease [60,62,63,66]. Further investigation showed that SSX proteins promote the unfolding and derepression of 1q12 PCH during replication [74] (Figure 1), which leads to segregation abnormalities during anaphase and generation of micronuclei. Our results demonstrate a novel mechanism for the generation of PCH-associated genomic instability in cancer cells that specifically links the functions of SSX and PcG proteins. Interestingly, depletion of PcG factors did not phenocopy SSX expression, suggesting that the structural modification of 1q12 PCH was a direct effect of SSX binding rather than being caused by the depletion of PcG factors from this chromatin domain. This further suggested that SSX molecules do not target PcG complexes directly, but may associate with PcG chromatin domains by binding to other factors or directly to the chromatin. By studying SSX2 deletion mutants, we showed that the SSXRD was essential for association with PcG bodies, whereas the KRAB domain was essential for structural modification of PcG bodies. In agreement with this, the SS18-SSX fusion protein has not been reported to deplete PcG bodies, although it associates with these structures similar to wild-type SSX [97,98]. From these results, we propose that SSX binds 1q12 PCH via the SSXRD and recruits factors that promote chromatin reorganization via the KRAB domain. In turn, this leads to the disintegration of PcG bodies and loss of PcG factors from 1q12 PCH [74]. Whether a similar interaction between SSX and PcG factors occurs in other PCH domains remains elusive. 

## 8. Future Directions 

Several questions remain unanswered in regard to the regulation and deregulation of PCH during tumorgenesis and the involvement of PcG and SSX proteins. It is important to understand what cancer cells gain from reprogramming PCH into PcG domains. Since PcG bodies are present in a large percentage of malignancies (e.g., 80% of melanomas) [52], this question should be a high priority. Moreover, attention should be directed to elucidating the molecular and cellular functions of SSX proteins. We are currently investigating the genome-wide binding profile of SSX, which should contribute to the overall clarification of the role of these proteins in genome regulation. It will be interesting to compare SSX deposition to that of PcG proteins and PcG-associated histone modifications to elucidate the functional collaboration of these factors. The specific role of SSX proteins in structurally modifying PcG-repressed heterochromatin and its role in tumor development should be another focus point, particularly the mechanism by which SSX proteins rearrange PcG-repressed PCH. Moreover, the chromatin-binding capacity of the SSXRD should be further investigated, and interaction partners of SSX proteins that have the potential to structurally modify chromatin should be identified. Collectively, this will contribute to our understanding of the implications of destabilization of PCH in cancer development and progression and potentially reveal novel therapeutic entry points.

## Figures and Tables

**Figure 1 cells-09-00226-f001:**
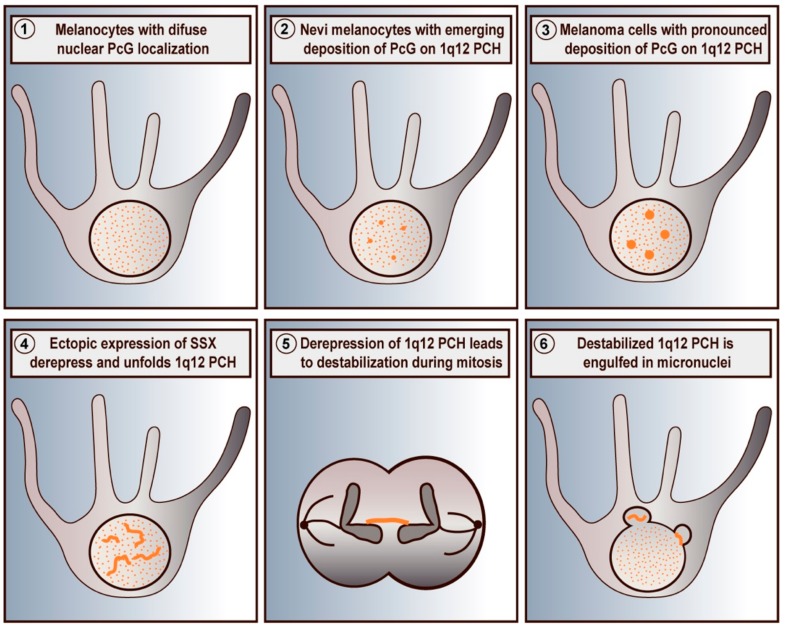
Schematic depiction of SSX-mediated destabilization of PcG-repressed 1q12 PCH domains in melanoma cells. Primary melanocytes exhibit a diffuse chromatin-associated distribution of PcG proteins (**1**). In melanocytes of nevi, PcG factors accumulate on 1q12 PCH (**2**) forming PcG bodies, which become more pronounced in melanoma cells (**3**). Ectopically expressed SSX proteins target and reorganize PcG bodies leading to the unfolding and derepression of 1q12 PCH domains (**4**). In turn, this destabilizes 1q12 PCH and results in formation of chromatin bridges during mitosis (**5**) and subsequent generation of micronuclei (**6**). Based on data from previous publications [52,74].

**Figure 2 cells-09-00226-f002:**
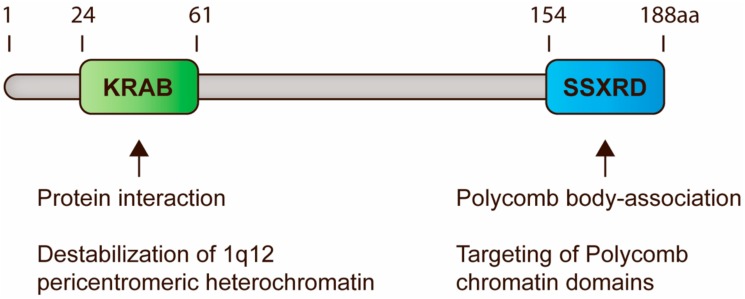
Schematic representation of SSX domains and their functions in regulation of PCH. All SSX proteins contain two highly conserved domains, i.e., the KRAB and SSXRD domains. KRAB domains are present in a large number of mammalian proteins and generally mediate protein interactions. In SSX molecules, this domain is essential for destabilization of 1q12 PCH, perhaps due to the recruitment of chromatin modifiers. The SSXRD domain structurally resembles C2H2 zinc finger motifs and is important for the targeting of SSX proteins to PcG-associated chromatin. Based on data from previous publications [74,88,97,98].

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
