# Peer review of "Interaction between Polycomb and SSX Proteins in Pericentromeric Heterochromatin Function and Its Implication in Cancer"

_cells, 2020, doi:10.3390/cells9010226_

Round 1
Reviewer 1 Report
In this manuscript, Johansen and Gjersttorff review the role of Polycomb and SSX proteins at pericentromeric heterochromatin. While Polycomb proteins are primarily known for their role in the regulation of stem cell differentiation and development, they are recruited to pericentromeric heterochromatin under specific circumstances, notably in malignant cells. In this context, the SSX family of proteins has emerged as Polycomb-interacting factors, which is discussed in the current review.
Overall, the review presents a timely topic that is highly relevant due to its potential implications for tumor biology. However, there are several points that should be considered/revised:
1) While the review discusses most relevant previous publications, there is a strong bias towards the author’s own data. In particular, the abstract reads more like a summary of the author’s work than a balanced literature review. The same applies to the SSX section of the manuscript.
2) The composition of PRC1 complexes, described on line 38, is incorrect. Canonical PRC1 complexes where reported to contain PCGF2/4, while variant PRC1 contains PCGF1/3/5/6 (see for example Gao et al, 2012; Schuttengruber et al, 2017).
3) In my opinion, the beginning of the second paragraph ‘The genome targeting of PRC complexes remains controversial.’ is rather outdated by now. While there are clear differences between Drosophila and mammals, quite some progress has been made in describing the process and the contributions of different entities (discussed for example in Piunti and Shilatifard, 2016; Schuttengruber et al, 2017).
4) Along the same lines, ‘the hierarchal [should read ‘hierarchical’] model of PRC recruitment’ (line 49) has been challenged multiple times. In fact, the authors refer to this in the next sentence, but the conclusion does not come across. Rather the statement ‘PcG recruitment … is more complex and complicated than assumed’ leaved the reader in confusion. In this context, the publication from Fursova et al, Mol Cell 2019 should be added.
5) The authors often loosely refer to ‘methylation’ without clarifying which type of methylation is discussed. In particular, in section 3 it is relevant whether DNA methylation, H3K9me3 or H3K27me3 is meant.
6) Lines 107-109, ‘transcriptional silencing, DNA methylation and chromatin compaction’ are no ‘factors’.
7) Lines 117-119, ‘Paternal PCH contains protamins at the … zygote stage and is therefore devoid of SUV39H-mediated compaction’ is an incorrect statement. Protamines are immediately exchanged to histones upon sperm decompaction. However, these histones initially lack H3K9me3 (see Santos et al, 2005; van der Heijden et al, 2006; Puschendorf et al, 2008).
8) The hypothesis in lines 125-127 has actually been tested (see Tardat et al, Mol Cell 2015). This should be included and the statement modified accordingly.
9) Line 157, H3K9me2/3 is not a factor.
10) Line 159, ‘Polycomb proteins have been demonstrated to deposit on hypomethylated DNA’. The data actually refers to hypomethylated CGIs (see the cited papers). Are the human pericentromeric regions discussed here CG-rich? In the mouse, the opposite is the case as minor satellites in the mouse are highly AT-rich. In this specific case, the Cbx2 AT-hook has been implicated in targeting (see Tardat et al, Mol Cell 2015). What is the situation for human satellite repeats?
11) Lines 166-176, what are the conclusions based on? I could not find any references.
12) Since the authors discuss Polycomb bodies at several occasions, I think it is appropriate to include the recent data on liquid-liquid phase separation (LLPS) and Polycomb proteins (see for example Illingworth, Current Op Genet Dev 2019; Tatavosian et al, J Biol Chem 2019; Plys et al, Genes Development 2019).
13) What does ‘SSX’ stand for?
14) PRC is misspelled on several occasions (‘PCR’). The same applies to PCH (‘PHC’).
15) Lines 295, what percentage of malignancies? Reference?
Author Response
Dear reviewer,
Thank you very much for considering our manuscript. We were delighted to see that our work was well received, and we greatly appreciate the opportunity to respond to the relevant and constructive criticism. We have thoroughly addressed the comments and we have included a point-by-point response below.
Comment 1: While the review discusses most relevant previous publications, there is a strong bias towards the author’s own data. In particular, the abstract reads more like a summary of the author’s work than a balanced literature review. The same applies to the SSX section of the manuscript.
Response: We acknowledge the reviewer’s opinion on this and has changed the wording of the abstract a bit to make it less biased towards our own data (line 20). We do not agree that the SSX section is biased towards our own research. The first and second parts (5. The SSX family of chromatin-modulating proteins and 6. Interaction between SSX and PcG factors) are a general introduction with references to work from many different research groups. The last part (7. SSX-mediated derepression of PcG-silenced heterochromatin) is mainly based on our own work, which is only natural since there is no data from others groups on this subject.
Comment 2: The composition of PRC1 complexes, described on line 38, is incorrect. Canonical PRC1 complexes where reported to contain PCGF2/4, while variant PRC1 contains PCGF1/3/5/6 (see for example Gao et al, 2012; Schuttengruber et al, 2017).
Response: Thank you for pointing this out. It’s been edited to: ‘’The four core subunits of canonical PRC1 (cPCR1) include an E3 ubiquitin ligase (RING1A/B), as well as proteins of the chromobox (CBX2/4/6/7/8), polyhomeotic-like protein (PHC1-3) and PcG group ring finger protein (PCGF2/4) families, respectively. In contrast, non-canonical PRC1 (ncPCR1) contain PCGF1/3/5/6 and lacks the CBX domain and instead possesses the zinc-finger domain containing RYBP or YAF2 protein [2, 9-11]’’ (line 36).
Comment 3: In my opinion, the beginning of the second paragraph ‘The genome targeting of PRC complexes remains controversial.’ is rather outdated by now. While there are clear differences between Drosophila and mammals, quite some progress has been made in describing the process and the contributions of different entities (discussed for example in Piunti and Shilatifard, 2016; Schuttengruber et al, 2017).
Response: We agree and has changed the paragraph accordingly (line 66).
Comment 4: Along the same lines, ‘the hierarchal [should read ‘hierarchical’] model of PRC recruitment’ (line 49) has been challenged multiple times. In fact, the authors refer to this in the next sentence, but the conclusion does not come across. Rather the statement ‘PcG recruitment … is more complex and complicated than assumed’ leaved the reader in confusion. In this context, the publication from Fursova et al, Mol Cell 2019 should be added.
Response: Agreed. It was not that clear in the text. And thanks for the publication from Fursova. Edited to: ‘’Initially, PRC recruitment was proposed to be mediated in a hierarchical manner, in which the deposition of H3K27me3 by PRC2 recruited cPRC1 [2, 14]. However, the combinatorial assembly of mammalian PRC1 subunit homologues gives rise to various functionally distinct ncPRC1 complexes that can occur and function independent of PRC2 and additionally can recruit PRC2 [11, 15-17], suggesting that PcG recruitment to chromatin and gene silencing is more complicated and complex than first assumed’’ (line 58).
Comment 5: The authors often loosely refer to ‘methylation’ without clarifying which type of methylation is discussed. In particular, in section 3 it is relevant whether DNA methylation, H3K9me3 or H3K27me3 is meant.
Response: We agree, and this has been corrected when appropriate.
Comment 6: Lines 107-109, ‘transcriptional silencing, DNA methylation and chromatin compaction’ are no ‘factors’.
Response: What is meant is that HP1 and methylated CGIs function as docking sites for additional factors that promote transcriptional silencing, DNA methylation and chromatin compaction, which are important for the formation of constitutive heterochromatin. The sentence has been corrected according to the comment and now reads: ‘’ HP1 and methylated CGIs both function as docking sites for additional factors that promotes transcriptional silencing, DNA methylation and chromatin compaction, which are important for the formation of constitutive heterochromatin [45-47]” (line 130).
Comment 7: Lines 117-119, ‘Paternal PCH contains protamins at the … zygote stage and is therefore devoid of SUV39H-mediated compaction’ is an incorrect statement. Protamines are immediately exchanged to histones upon sperm decompaction. However, these histones initially lack H3K9me3 (see Santos et al, 2005; van der Heijden et al, 2006; Puschendorf et al, 2008).
Response: Thanks for the clarification. The sentence has been corrected and now reads: ‘’To the contrary, paternal PCH is initially devoid of H3K9me3, and is depend on the incorporation of maternal PRC1 and the subsequent recruitment of PRC2 and H3K27me3 deposition to ensure chromatin compaction’’ (line 140).
Comment 8: The hypothesis in lines 125-127 has actually been tested (see Tardat et al, Mol Cell 2015). This should be included and the statement modified accordingly.
Response: Thanks for the publication. The suggested reference has been added to support the earlier statement and the sentence now reads: ‘’To the contrary, paternal PCH is initially devoid of SUV39h-mediated H3K9me3, and is depend on the incorporation of maternal PRC1 and the subsequent recruitment of PRC2 and H3K27me3 deposition to ensure chromatin compaction’’ (line 140), as well as the, now corrected, statement: “Several studies have investigated the relationship between H3K9me3, CGI methylation and PcG recruitment to PCH in human and mice embryonic stem cells [16, 42-44, 54], and found that PRC2 recruitment correlates with loss of methylation. In this regard, it has been suggested that H3K9me3/HP1 and methylated CGI antagonize PRC recruitment, and thus inhibition of SUV39H DNMT and complete loss of H3K9me3/HP1 and CGI methylation allows PRC recruitment and activity’’ (line 140).
Comment 9: Line 157, H3K9me2/3 is not a factor.
Response: We agree and have corrected this (line 125).
Comment 10: Line 159, “Polycomb proteins have been demonstrated to deposit on hypomethylated DNA”. The data actually refers to hypomethylated CGIs (see the cited papers). Are the human pericentromeric regions discussed here CG-rich? In the mouse, the opposite is the case as minor satellites in the mouse are highly AT-rich. In this specific case, the Cbx2 AT-hook has been implicated in targeting (see Tardat et al, Mol Cell 2015). What is the situation for human satellite repeats?
Response: We agree that this has been imprecisely presented in the manuscript and have corrected the paragraph accordingly (Line 208).
Comment 11: Lines 166-176, what are the conclusions based on? I could not find any references.
Response: We agree that this paragraph lacks references. This has been corrected (line 212-224).
Comment 12: Since the authors discuss Polycomb bodies at several occasions, I think it is appropriate to include the recent data on liquid-liquid phase separation (LLPS) and Polycomb proteins (see for example Illingworth, Current Op Genet Dev 2019; Tatavosian et al, J Biol Chem 2019; Plys et al, Genes Development 2019).
Response: We agree, and have briefly included these data in the manuscript (line 195).
Comment 13: What does ‘SSX’ stand for?
Response: The full name of SSX is now presented in the manuscript (line 245).
Comment 14: PRC is misspelled on several occasions (‘PCR’). The same applies to PCH (‘PHC’).
Response: This has been corrected.
Comment 15: Lines 295, what percentage of malignancies? Reference?
Response: As an example, we have included the number of melanoma tumors with PcG bodies and provided the reference for the data (line 359).
Reviewer 2 Report
This is an interesting, well-written review that covers an important axis, Polycomb and SSX proteins, in genome stability and cancer. The work provides a comprehensive summary regarding the main players (polycomb group complexes, SSX proteins), their roles in genome stability and cancer and is a good synopsis of some of the recent work from the authors' lab regarding this topic. Thus, this review covers an important niche.
I have only a few comments:
(1) I think the authors could raise more attention, a greater readership by integrating the cancer angle into their title. Maybe something like this:
"Interaction between Polycomb and SSX proteins in pericentromeric heterochromatin function and its implication in cancer".
The title as it stands now is very much matter of fact, but maybe a bit boring and does not elude to the important cancer implications.
(2) Page 3, line 116, should it not be H4k20me3 (not H4K30me3)?
(3) Page 4, lines 171 etc "..and we have demonstrated that PcG bodies emerge in SV40ER- and hTERT- immortalized melanocytes". Could you please provide the references here?
(4) Fig 1. legend"..nevi PcG, factors.." a comma too much...I thjink remove comma between nevi and PcG.
(5) Page 5, line 186: "The SSX family comprises 6 highly identical members.."
either 'identical' or 'highly similar' or 'near identical', but not 'highly identical', please.
(6) Page 6, line 221-222:"similar to our results..", please, provide reference here or highlight ahead what results came from your lab.
(7) Page 7, line 281: "..SSX molecules does not.." should be "..SSX molecules do not..".
Author Response
Thank you very much for considering our manuscript. We were delighted to see that our work was well received, and we greatly appreciate the opportunity to respond to the relevant and constructive criticism. We have thoroughly addressed the comments and we have included a point-by-point response below.
Comment 1: I think the authors could raise more attention, a greater readership by integrating the cancer angle into their title. Maybe something like this:
"Interaction between Polycomb and SSX proteins in pericentromeric heterochromatin function and its implication in cancer".
The title as it stands now is very much matter of fact, but maybe a bit boring and does not elude to the important cancer implications.
Response: We agree and have changed the title accordingly.
Comment 2: Page 3, line 116, should it not be H4k20me3 (not H4K30me3)?
Response: This indeed correct and has been changed (line 139).
Comment 3: Page 4, lines 171 etc "..and we have demonstrated that PcG bodies emerge in SV40ER- and hTERT- immortalized melanocytes". Could you please provide the references here?
Response: As suggested, a reference has been included (line 219).
Comment 4: Fig 1. legend"..nevi PcG, factors.." a comma too much...I thjink remove comma between nevi and PcG.
Response: This has been corrected as suggested.
Comment 5: Page 5, line 186: "The SSX family comprises 6 highly identical members.."
either 'identical' or 'highly similar' or 'near identical', but not 'highly identical', please.
Response: This has been corrected as suggested (line 245).
Comment 6: Page 6, line 221-222:"similar to our results..", please, provide reference here or highlight ahead what results came from your lab.
Response: We have clarified where “our results” are presented and included a reference (line 282).
Comment 7: Page 7, line 281: "..SSX molecules does not.." should be "..SSX molecules do not..".
Response: This has been corrected.
Round 2
Reviewer 1 Report
The authors have addressed all comments that were raised, and I therefore recommend the manuscript for publication.